# Suitability of Gelatin Methacrylate and Hydroxyapatite Hydrogels for 3D-Bioprinted Bone Tissue

**DOI:** 10.3390/ma17051218

**Published:** 2024-03-06

**Authors:** Paul Stolarov, Jonathan de Vries, Sean Stapleton, Lauren Morris, Kari Martyniak, Thomas J. Kean

**Affiliations:** Biionix Cluster, Department of Internal Medicine, College of Medicine, University of Central Florida, Orlando, FL 32827, USAse898126@ucf.edu (S.S.);

**Keywords:** 3D bioprinting, hydroxyapatite, GelMA, bone, design of experiments

## Abstract

Background: Complex bone defects are challenging to treat. Autografting is the gold standard for regenerating bone defects; however, its limitations include donor-site morbidity and increased surgical complexity. Advancements in 3D bioprinting (3DBP) offer a promising alternative for viable bone grafts. In this experiment, gels composed of varying levels of gelatin methacrylate (GelMA) and hydroxyapatite (HA) and gelatin concentrations are explored. The objective was to increase the hydroxyapatite content and find the upper limit before the printability was compromised and determine its effect on the mechanical properties and cell viability. Methods: Design of Experiments (DoE) was used to design 13 hydrogel bioinks of various GelMA/HA concentrations. These bioinks were assessed in terms of their pipettability and equilibrium modulus. An optimal bioink was designed using the DoE data to produce the greatest stiffness while still being pipettable. Three bioinks, one with the DoE-designed maximal stiffness, one with the experimentally defined maximal stiffness, and a literature-based control, were then printed using a 3D bioprinter and assessed for print fidelity. The resulting hydrogels were combined with human bone-marrow-derived mesenchymal stromal cells (hMSCs) and evaluated for cell viability. Results: The DoE ANOVA analysis indicated that the augmented three-level factorial design model used was a good fit (*p* < 0.0001). Using the model, DoE correctly predicted that a composite hydrogel consisting of 12.3% GelMA, 15.7% HA, and 2% gelatin would produce the maximum equilibrium modulus while still being pipettable. The hydrogel with the most optimal print fidelity was 10% GelMA, 2% HA, and 5% gelatin. There were no significant differences in the cell viability within the hydrogels from day 2 to day 7 (*p* > 0.05). There was, however, a significantly lower cell viability in the gel composed of 12.3% GelMA, 15.7% HA, and 2% gelatin compared to the other gels with a lower HA concentration (*p* < 0.05), showing that a higher HA content or print pressure may be cytotoxic within hydrogels. Conclusions: Extrusion-based 3DBP offers significant advantages for bone–tissue implants due to its high customizability. This study demonstrates that it is possible to create printable bone-like grafts from GelMA and HA with an increased HA content, favorable mechanical properties (145 kPa), and a greater than 80% cell viability.

## 1. Introduction

Bone is versatile tissue that has a significant capacity for regeneration. However, the successful regeneration of complex bone defects is currently one of the biggest challenges faced by reconstructive surgeons [1]. Smaller bone defects can usually heal without intervention, as is the case with many types of fractures and small cranial bone defects [2]. Large or complex defects, termed critical-sized defects, exceed the natural capacity for regeneration, requiring further intervention [2]. Currently, the most common methods used for repairing large bone defects are autografting (transplanting bone from another site in a single patient) and allografting (transplanting cadaver bone) [2]. These structures are often hand-carved to fit the size of the defect [2]. Despite being considered the current gold standard, autografting and allografting are associated with significant drawbacks. Autografting can lead to donor-site morbidity and requires more complex surgery. Allografting may be limited by the supply of donors, the potential for immunogenicity, and the risk of disease transmission [1]. Although their conflicts of interest should be noted, studies on cell-infused cellular bone matrices have shown high success rates and low complication rates in spinal fusion [3].

In the past decade, tissue engineering of bone using Three-Dimensional Bioprinting (3DBP) has emerged as a possible solution to the challenges associated with bone autografting and allografting [4]. Three-Dimensional Bioprinting is the fabrication of a tissue-like construct through the deposition of living cells, often suspended in a hydrogel. When the cells and the hydrogel are combined, they make up a liquid “bioink” that can be extruded from a bioprinter in layers to produce a cellular scaffold [5,6]. This scaffold can be subjected to covalent photocrosslinking during or after printing, producing a hardened construct that resembles the bone microenvironment. A significant advantage of 3DBP over allografting and autografting is that patient-derived autologous mesenchymal stromal cells (hMSCs) can be used in the construct, reducing the potential for graft immunogenicity and donor-site morbidity. Additional advantages of 3DBP over allografting are the decreased risk of infection, the ability to easily create large complex shapes, and the use of organic and biological additives to aid osteogenesis [5,7].

A significant challenge in 3DBP bone-like constructs is formulating a bioink that closely mimics the bone microenvironment. Natural bone is typically 30% organic material (collagen), 60% mineral, and 10% water [8]. The Young’s modulus of cortical and trabecular bone is ~19.9 GPa and ~18.0 GPa, respectively [9]. Meanwhile, the modulus of the pre-calcified bone matrix housing the bone marrow is around 64 kPa [10]. The bone bioinks currently used for 3DBP generally consist of three components: the base material, living cells, and various natural or synthetic additives. The base materials (most commonly alginate, gelatin methacrylate, or hyaluronic acid) in the bioink can be crosslinked after printing to increase the mechanical stiffness of the construct significantly [7,11].

Gelatin (a hydrolyzed product of collagen) with methacryloyl side groups (GelMA) is a popular base material in bone 3DBP because of its ability to be crosslinked in the presence of a photoinitiator [12,13,14]. Low concentrations of pure GelMA (5% *w*/*v*) can be extruded more easily and generate a higher cell viability [15]. However, low concentrations of GelMA lack the stiffness needed to produce a construct that resembles the rigidity and load-bearing capacity of bone. Higher concentrations of GelMA (15% *w*/*v*) require higher pressure to be extruded and may decrease the cell viability, although this can be overcome by warming the material to become more liquid before extrusion [15]. To maximize the printability and cell viability, most groups have used GelMA concentrations between 5% and 15% *w*/*v* for 3DBP [7]. In addition, low concentrations of pure gelatin can be added to GelMA hydrogels to increase the viscosity and printability of the material without significantly affecting their material characteristics [16].

Natural and synthetic additives such as hydroxyapatite, bioactive glass, and graphene oxide have a significant impact on the structural stability of bioink [17]. Additives can also increase the cell viability within the hydrogel in low concentrations and promote differentiation into target tissues like bone or cartilage [18]. Hydroxyapatite (HA) is the primary inorganic material found in bone tissue, which makes it a desirable additive for bone tissue engineering [17]. When used as an additive with pure gelatin hydrogels, HA has been shown to improve the mechanical stiffness of engineered bone tissue by up to 70% (42 kPa to 72 kPa) [19]. However, higher concentrations of HA may decrease the cell viability due to its high calcium content [20].

Some of the most important characteristics of a bioprinted structure are printability, print fidelity, mechanical strength, and cell viability [21]. Printability, also called extrudability in extrusion bioprinting, is defined as the printing parameters necessary to print a straight line of a defined width. Printability is influenced by pressure, temperature, speed, acceleration, and the diameter of the needle used for printing [21]. Determining printability is an important step in evaluating the feasibility of printing a bioink. Print fidelity is the degree to which the bioprinted structure holds its shape, and it is related to the viscosity of the bioink [22]. Three-dimensional bioprinted structures often require a high print fidelity, and it is crucial to balance the print fidelity with the printability to enable custom, defect-specific implants based on high-resolution clinical imaging [7]. The mechanical stiffness of the printed product is particularly important, as bioprinted bone structures should have significant load-bearing capabilities to maximize the bone healing when implanted [23]. A material’s mechanical properties can be determined by stress-testing it. In this study, we will calculate the equilibrium modulus to determine the mechanical properties of the materials [24]. Lastly, the high pressures and cross-linking procedures required for 3DBP bone tissues with a high mechanical strength can have a deleterious effect on cell viability [15]. Evaluating the viability of the human cells in each printable bioink after the printing process is an important step in determining their suitability as bone replacement implants.

A previous study used GelMA and HA for 3DBP and determined that the addition of HA in low concentrations (≤2%) to a 10% GelMA hydrogel maintained equal cell viability and decreased the hydrogel swelling in comparison to constructs without HA [25]. The ideal combination of these materials for optimizing the printability, print fidelity, mechanical strength, and cell viability remains largely unknown. 

Testing bioprinting with a wide range of material concentrations using more than one material is challenging. The Design of Experiments software (DoE; Design-Expert, Stat-Ease, version 22.0.2, Minneapolis, MN, USA) can aid in the experimental planning and analysis [13,14]. The DoE software allows researchers to determine the optimal values for further testing using fewer experiments based on the desired outcomes [26].

For example, if the DoE software is given the maximum and minimum values of two material concentrations within a hydrogel, it will determine a subset of combinations within that range that will produce the greatest coverage of the design space with the fewest runs in an experiment. After the experiment, the results can be analyzed using the DoE software, which can then predict the ideal concentrations of both materials in a hydrogel for a desired outcome [14].

The objective of this study was therefore to determine the ideal GelMA/HA bioink combination with added gelatin for extrusion-based 3DBP of bone-like constructs laden with hMSCs. To accomplish this, we experimentally evaluated the effects of various combinations of GelMA and HA with 2% gelatin on printability, print fidelity, mechanical properties, and cell viability. 

We hypothesized there would be a direct relationship between the GelMA/HA concentrations and the equilibrium modulus but an inverse relationship between the GelMA/HA concentrations and printability, print fidelity, and cell viability. 

## 2. Materials and Methods

### 2.1. Cell Culture

Human bone-marrow-derived mesenchymal stromal cells (hMSCs) were thawed at passage 2 from frozen stock of hMSCs isolated under the Institutional-Review-Board-approved non-human research protocols (University of Central Florida College of Medicine STUDY00001124) and cultured until passage 5, at which point the experiments were performed. The cells were cultured in growth medium containing Dulbecco’s Modified Eagle Medium (DMEM; Gibco Inc., Grand Island, NY, USA) supplemented with 10% fetal bovine serum (FBS, Atlanta Biologicals Inc., Flowery Branch, GA, USA), and 1% penicillin/streptomycin (Gibco). The cells were incubated in a humidified environment at 37 °C with 5% CO_2_. At ~90% confluence, the cells were trypsinized (0.25% trypsin/EDTA, Gibco), neutralized with growth media, centrifuged (500 RCF, 5 min), and resuspended in growth media. The live cells were counted (CytoSMART, Atlanta, GA, USA) using trypan blue exclusion. Cells were combined with the bioinks at a final concentration of ~1 × 10^6^ cells/mL.

### 2.2. Design of Experiments Parameters

Experimental design software (Design-Expert, Stat-Ease Inc.) was utilized to determine the hydrogel formulations for testing based on GelMA concentrations between 5% and 15% *w*/*v* and HA concentrations between 1% and 33%. The DoE software produced 17 experimental runs with various GelMA/HA bioink formulations at 2% gelatin, some of which were repeated to test the experimental variability (Table 1).

The 17 hydrogels were first assessed based on their pipettability and the equilibrium modulus of a cast cylinder. The bioinks with ideal printability and mechanical properties were later 3D-bioprinted and evaluated for their print fidelity and cell viability. 

### 2.3. Bioink Formulation

The 17 experimental bioinks (Table 1) were prepared using GelMA at a methacrylation level of 45–55% (Rousselot Inc., Gent, Belgium) in PBS containing 0.05% lithium phenyl-2,4,6-trimethyl-benzoyl phosphinate (LAP) as the photoinitiator (CELLINK Inc., San Carlos, CA, USA) with 2% 300 bloom gelatin (Electron Microscopy Sciences Inc., Hatfield, PA, USA). The solution was warmed at 37 °C on a hot plate and vortexed at 1500 revolutions per minute until dissolved. Various concentrations of HA powder (Sigma-Aldrich, St. Louis, MO, USA) were added to the solution and stirred until suspended. The hydrogel was combined with the cultured hMSCs at a concentration of ~1 × 10^6^ cells/mL [27].

### 2.4. Printability

For the purposes of this study, printability was defined as the ability of a material to be pipetted and extruded as a gel at a temperature under 37 °C. A 0.2 mL sample of each of the 17 composite hydrogels (no cells) was warmed to 37 °C and pipetted. Bioinks that could not be pipetted or extruded were deemed unprintable. A composite score was developed for each bioink from the results, where printability is a dichotomous variable (1 = printable, 0 = not printable). 

### 2.5. Mechanical Testing

The mechanical testing was performed using a TA.XTPlusC mechanical testing device (Stable Micro Systems Inc., Hamilton, MA, USA). Each of the 17 acellular composite hydrogels were heated to 37 °C and pipetted into circular casts made using a biopsy punch on a silicone mat, giving discs 5 mm in diameter by 0.5 mm in height. The discs and casts were held using a microscope slide and coverslip. They were then photocrosslinked using a 300 mW laser with visible light at 405 nm (Luck Laser Inc., Hangzhou, China). The laser was positioned 3.5 cm above the hydrogels, which were treated on both sides for 60 s. 

A Texture Analyzer 12.7 mm diameter cylindrical probe (TA-10) with a 5 kg load cell was used for compression. Uniaxial compression was initiated at a speed of 0.1 mm/s and a trigger force of 0.1 g. The cylinders were compressed to 5, 10, 15, 20 and 25% strain and allowed to equilibrate for 5, 10, 15, 30 and 45 min, respectively. The materials were tested at room temperature in PBS. The output was plotted using the Exponent Connect software (Version 8.0.16.0, Stable Micro Systems Inc.) to produce the force/time curve. The slope of the curve fitted to those stress/strain points was taken as the equilibrium modulus.

### 2.6. Bioprinting 

The bioprinting was performed using a BioAssemblyBot 3D bioprinter (Advanced Solutions Inc., Louisville, KY, USA) and using pneumatic extrusion bioprinting (Figure 1). Each bioink was loaded into a disposable UV-blocking amber cartridge (Nordson Inc., Swainsboro, GA, USA) with 20-gauge 0.5-inch straight needle dispensing tips (Nordson Inc.). The Tissue Structure Information Modeling (TSIM, Version 1.1.227, Advanced Solutions Inc.) software was used for the 3D modeling. After printing, the constructs were photocrosslinked using a 200 mW laser and visible light at 405 nm positioned 3.5 cm above the construct (Luck Laser Inc.). 

### 2.7. Development of a Laser Adapter on the Hot Tool 

When we started these experiments, the hot tool did not have the ability to photcrosslink. A separate tool could be attached that would crosslink at 365 nm. We therefore sought to develop an on-tool laser to enable photocrosslinking while printing. A 405 nm laser (200 mW, Luck Laser) was chosen, as this wavelength does less damage to DNA than the 365 nm wavelength [28] and can photocrosslink using the LAP photoinitiator [13,14]. A holder arm was 3D-printed in PLA to attach the laser, battery packs, and Raspberry Pi Pico (Figure 1D) [27]. This enables the printing of thicker constructs, as crosslinking may not be possible after the print is completed.

### 2.8. Print Settings

The ideal printing pressures and temperatures for the composite hydrogels were determined using a systematic approach where the pressure was gradually increased in increments of 5 psi, while the temperature was increased from 30 °C to 37 °C at each interval until a continuous filament was achieved.

### 2.9. Print Fidelity

Print fidelity is defined as the degree to which a bioprinted structure holds its shape [22]. The bioinks were bioprinted into a square zig-zag pattern (Figure 2) [13]. The constructs were crosslinked using the above-mentioned procedure and imaged (VHX-7000, Keyence Inc., Tampa, FL, USA).

Photographs were taken and overlayed onto the TSIM model. The area was measured using an area calculator [29]. These measurements were used to calculate the print fidelity according to the equation below, where ***A_th_*** is the theoretical pore area and ***A_e_*** is the experimental pore area. A value of 100% is a perfect print fidelity, a value below 100% indicates underprinting, and a value above 100% represents overprinting.
(1)$$Print\;Fidelity\left(\%\right)=\frac{A_{th}}{A_{e}}\;\times \;100$$

### 2.10. Viability

Viability testing was performed using the LIVE/DEAD Viability/Cytotoxicity Kit (Thermo Fisher Inc., Waltham, MA, USA). The bioinks were both pipetted and printed onto a well plate in triplicate. The bioinks were crosslinked and incubated in growth medium for 2 days and 7 days at 37 °C and 5% CO_2_. After each time point, the growth medium was removed, and the constructs were washed in PBS. A solution containing 2 µM of Calcein AM (Invitrogen, Carlsbad, CA, USA) and 4 µM of ethidium homodimer-1 (Invitrogen) in PBS was added to the wells and incubated for 25 min at 37 °C and 5% CO_2_. The dye was then removed, and PBS was added to the wells. The constructs were then imaged and overlayed (Keyence VHX-7000 microscope). The images were processed and analyzed using the ImageJ/Fiji software (Version 1.52p, NIH) [30] and using the StarDist plugin [31].

### 2.11. Statistical Analysis

The design space covered by the DoE combinations was analyzed (Stat-Ease). The z-axis is the standard error of design, and the x/y-axes are the GelMA/HA concentrations as pictured (Figure 3). Following testing, ANOVA was performed using the DoE software to demonstrate the effects of the GelMA and HA concentrations on the equilibrium modulus and printability and the overall fit of the data to the model. Based on the compression and printability results, we used the DoE software to determine the ideal concentrations of GelMA and HA to optimize the mechanical strength. The constraints used were that GelMA had to lie between 5 and 15%, HA between 1 and 33%, the pipettability was targeted to be =0.999 (as close to 1 as possible) with a lower limit of 0.8, and the goal was to maximize the equilibrium modulus with a lower limit of 160 kPa and an upper limit of 354 kPa. 

The statistical analysis and data illustration for cell viability were performed using the GraphPad Prism software (Version 10.1.1, GraphPad, Boston, MA, USA). A two-way ANOVA test followed by Sidak’s correction for multiple comparisons in a post hoc test was used to determine statistical significance. Where applicable, data are shown as mean +/− standard deviation. 

## 3. Results

### 3.1. Printability of Hydrogels

Of the 13 different hydrogels examined, 4 were unprintable. G10H33, G12.5H25, G15H17, and G15H33 had chalky appearances and did not stay in suspension well. None could be homogenously pipetted at 37 °C and were therefore deemed not printable (Figure 4).

### 3.2. Mechanical Properties

The gel replicates that were tested showed little variability int their mechanical properties (e.g., G10H17 mean 123.6 ± 5.5 kPa). The mechanical testing showed a positive correlation between both the GelMA and HA concentrations and the equilibrium modulus (Figure 5).

The DoE ANOVA analysis indicated that the augmented three-level factorial design used was a good fit (*p* < 0.0001). GelMA, HA, and the combination of the two were significant factors affecting the equilibrium modulus (*p* < 0.0001). Using those data, we used the DoE software to determine that a composite hydrogel consisting of 12.3% GelMA, 15.7% HA, and 2% gelatin (Gel 1) would produce the maximum equilibrium modulus, predicted to be 170 kPa (Table 2). Because Gel 2 was the stiffest pipettable gel before DoE optimization, it was also chosen for further testing. Gel 3 was the bioink used by Allen et al. with the highest HA concentration (Table 2). Moving forward, we used the DoE predicted ideal (Gel 1), the best screened gel (Gel 2), and a literature-based control (Gel 3). 

### 3.3. Print Settings and Print Fidelity

#### 3.3.1. Print Settings

The ideal extrusion pressure and temperature for the three composite hydrogels were determined qualitatively. The print setup was essentially the same for all three gels (Figure 6A). The collection plate was heated to 35 °C. All the gels produced a continuous filament at two temperature/pressure settings (Figure 6B). As expected, higher temperatures and pressures were required to produce a continuous filament in the hydrogels with higher GelMA and HA concentrations (Figure 6B). An increased printing pressure or temperature beyond this point resulted in over-extrusion in all three hydrogels.

#### 3.3.2. Print Fidelity

With an increasing hydroxyapatite concentration, the materials qualitatively increased in opacity (Figure 6C). All three gels were overprinted compared to the CAD model. While no direct relationship between the print fidelity and GelMA or HA concentrations was observed, Gel 3 produced the best print fidelity at 155.23% (Figure 6C). Gel 2 produced a pattern with the most over-extrusion, with a print fidelity of 234.14%. It is predicted that print fidelity is more a reflection of the ideal print settings rather than the composition of the gel. The more ideal and precise the settings, the higher the print fidelity.

### 3.4. Viability

All the gels showed acceptable cell viability. There was a significantly higher cell viability in pipetted Gels 2 and 3 when compared to Gel 1 both pipetted and printed (*p* < 0.05). The LIVE/DEAD staining demonstrated that the higher HA concentrations or higher pressures needed to extrude Gel 1 may be cytotoxic to hMSCs (Figure 7A). In addition, there was no significant difference between the same hydrogels at different times (*p* > 0.05).

## 4. Discussion 

An initial screen of the hydrogel mixtures was used to make a DoE prediction for the stiffest printable gel. We successfully achieved a well-correlated prediction with Gel 1, composed of 12.3% GelMA, 15.7% hydroxyapatite, and 2% gelatin, giving an equilibrium modulus of ~170 kPa vs. the predicted 170 kPa. Gel 2, composed of 12.5% GelMA, 9% hydroxyapatite, and 2% gelatin, had a slightly lower equilibrium modulus (~146 kPa) but yielded significantly higher cell viability (~82%) compared to Gel 1 (~60%). Gel 3, composed of 10% GelMA, 2% hydroxyapatite, and 5% gelatin, had the highest print fidelity. Thus, we achieved an increase in the mineral content of bioink from 2% to 9% HA in Gel 3 with good viability. However, native bone has a mineral content of about 60% [8], so we fell significantly short of that target. It is expected that this scaffold would increase in mineralization with time either in culture or in vivo [31]. The use of DoE in combination with high concentrations of HA has not been reported before, and its results in terms of the bioink produced are on the higher end of the moduli exhibited by bioinks [7].

The cell viability decreases at temperatures above 37 °C; thus, a cutoff of 37 °C was chosen for printability. If a gel is not printable at 37 °C or below, it cannot sustain cell viability. It was also found that at temperatures above 37 °C, the hydrogels slowly weakened (i.e., lose compressive moduli), capping the max temperature at 37 °C.

There are multiple possible explanations for the drop in printability at high HA concentrations. For example, high amounts of HA may not have remained suspended within the hydrogels, which will have then formed a plug at the bottom of the syringe, preventing extrusion. 

Bone has the greatest stiffness of any tissue in the human body and can sustain immense amounts of pressure. As previously mentioned, the Young’s modulus of cortical and trabecular bone is ~19.9 GPa and ~18.0 GPa, respectively [9]. While this number may be impossible to achieve using hydrogels, a 3D-bioprinted bone construct with an equilibrium modulus > 64 kPa was achieved, making it similar to the hardness of the pre-calcified bone matrix [10]. In addition, the hMSCs may increase the hardness of the material over time due to the release of osteogenic molecules that increase its mechanical stiffness [29].

The increasing opacity of a high % of HA may have also affected the mechanical stiffness by preventing the laser from fully penetrating the material, creating non-uniform crosslinking. This was addressed by simultaneously printing and crosslinking. One drawback to this is that each part of the gel is only crosslinked for a moment as the laser passes through.

The two print settings that were quantified were temperature and pressure (Figure 6B). Both had a significant effect on the quality of the extruded filament. As expected, hydrogels containing higher GelMA and HA concentrations required a higher temperature and pressure to achieve a continuous filament. One of the challenges when heating the gel within the printer was a potential lack of uniformity of the temperature within the tube itself. This resulted in regions of varying temperature while printing, resulting in some patterns of over-extrusion followed by under-extrusion. The temperature near the tip of the heated syringe was also likely lower, as it was outside of the heating apparatus. As a result, higher pressures were needed in order to start extrusion and then the pressure would be lowered to maintain the print. Other factors that were less optimized were the print speed and bed temperature, which can have a significant effect on the fidelity. 

Print fidelity is crucial for assessing the ability of a bioink to be engineered for patient-specific implants and depends on many factors such as gelation time, temperature, print speed, and pressure. Bioinks with a high print fidelity will produce a value close to 100%, meaning they hold their shape to a significant degree. Bioinks with greater than 100% print fidelity will be overprinted and collapse on the central pore. Bioinks with lower than 100% print fidelity will be underprinted and produce a central pore area that is larger than that of the model. 

With the print settings shown in Figure 6B, Gel 3 produced the most optimal print fidelity. Although all three gels produced constructs that qualitatively modeled the computer-aided design, all were slightly overprinted. While some of the print fidelity results can be attributed to the characteristics of the gel, it is difficult to determine the degree to which the bioink itself influences the print fidelity. The material concentrations influence the time required for the material to become a solid after printing. However, this effect could be controlled by performing photocrosslinking during the bioprinting process. The gelation time, temperature, print speed and pressure can all heavily influence the print fidelity. It is likely that with more precise control over the temperature and pressure and with immediate photocrosslinking, all three gels could produce a more optimal print fidelity. Therefore, although Gel 3 produced the print fidelity closest to 100%, this may not mean that it is significantly better at modeling the computer-aided design than Gel 1 or Gel 2. Despite these limitations, we determined that Gel 3 performed the best with the print settings shown in Figure 6B. 

All three materials sustained adequate cell viability, indicating that hMSCs can survive within the hydrogels throughout the photocrosslinking process. The results also showed that all three hydrogels can sustain hMSC viability for at least 7 days in culture. Increasing concentrations of HA may have affected the cell viability in Gel 1, which had an HA concentration of 15.7%. This is significantly higher than the highest HA concentration previously tested in the literature, which was 2% HA [16]. We predicted that printed gel may have lower cell viability as a result of high extrusion pressures. This, however, was not the case, as the cell viability in Gel 1 printed was not significantly different from Gel 1 pipetted. The cell viability did not change with intermittent pressures of up to 40 psi. Of the three gels tested, Gel 2 seemed to exhibit a trend toward consistently higher viability, but this difference was not statistically significant when compared to Gel 3. These results indicate that although all three hydrogels are suitable for sustaining hMSCs for an extended period, Gel 2 may be the most optimal given the combination of stiffness and viability for 3D-bioprinted bone implants.

The cell viability in Gel 1 when printed did not change significantly when compared to Gel 1 pipetted. Analysis of the viability in Gel 2 and Gel 3 when printed was not possible as a result of the poor staining/high background, where individual cells could not be visualized (Appendix A). Given that the higher pressure necessary to extrude Gel 1 (20 psi vs. 14–18 psi for Gels 2 and 3) did not seem to significantly affect the viability, the cell viability likely would not have been different when comparing Gels 2 and 3 printed and pipetted.

The use of GelMa, HA, and gelatin is not the only combination being explored for hydrogels to 3D bioprint bone constructs. Although these materials are the most studied, there are also papers exploring hyaluronic-acid-derived, agarose, alginate-derived, cellulose-derived, and PEG-derived gels, among others [7]. Further, combining solid components with hydrogels offers intriguing possibilities, but discrepancies in the mechanical strength, particularly at the interface, can create weak planes prone to fracture, hindering the functionality of these composites [7].

## 5. Conclusions

Three-dimensional bone printing is a promising alternative to the current gold standard for the regeneration of critical-sized bone defects. The ideal bioink to produce bone-like tissue should be one that is easily printable and maintains adequate mechanical strength while sustaining cell viability for in vivo ossification. While the ideal materials for this bioink have yet to be discovered, the results of this study demonstrate that hydrogels combining GelMA and HA are suitable materials for the 3DBP of bone-like constructs and have promising future implications as 3D-bioprinted bone substitutes. This research showed that a GelMA/HA bioink consisting of 12.3% GelMA, 15.7% HA, and 2% gelatin (Gel 1) would produce the greatest mechanical stiffness, and a hydrogel consisting of 10% GelMA, 2% HA, and 5% gelatin (Gel 3) produced the best print fidelity. A gel composed of 12.5% GelMa, 9% HA, and 2% gelatin (Gel 2), however, would likely be the most ideal candidate for further research because of its high equilibrium modulus and its increased viability when compared to Gel 1. However, more research is required to determine whether GelMA/HA constructs can match the hardness of physiological bone after time in culture and whether the constructs promote the regeneration of critical-sized defects in animal models. Bioinks containing these concentrations of GelMA and HA should be considered in the development of future 3DBP bone implants for the reconstruction of complex defects. While we found utility in the binary system delineating pipettable and non-pipettable solutions, a full characterization of the rheological properties would be useful. This is a limitation of the current study and could be further expanded upon in future studies. Typical bioinks exhibit shear-thinning behavior, so determination of the exact point at which it breaks down within these composite materials would be beneficial.

## Figures and Tables

**Figure 1 materials-17-01218-f001:**
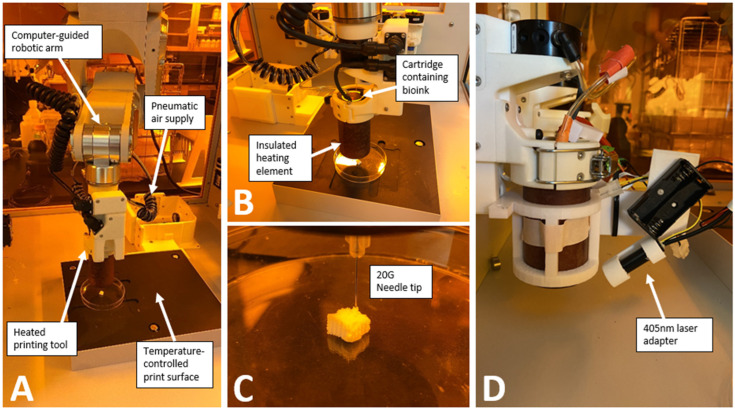
**Bioprinter assembly**: (**A**) BioAssemblyBot 3D bioprinter from Advanced Solutions Inc. (**B**) Cartridges containing the bioinks were inserted into the heating tool and allowed to warm to the desired temperature before printing. (**C**) Example of 3D construct printed with hydroxyapatite-rich composite hydrogel. (**D**) 405 nm laser adapter for simultaneous printing and crosslinking.

**Figure 2 materials-17-01218-f002:**
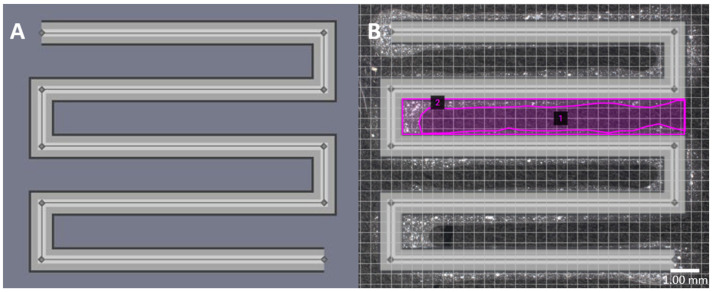
**CAD model used for print fidelity**: (**A**) Square zig-zag pattern of 8.9 mm length, 10.9 mm width, and line thickness of ~0.9 mm using TSIM. (**B**) Example of an overlaid image with areas of the actual print (Line 1) and theoretical pore size (Line 2) delineated.

**Figure 3 materials-17-01218-f003:**
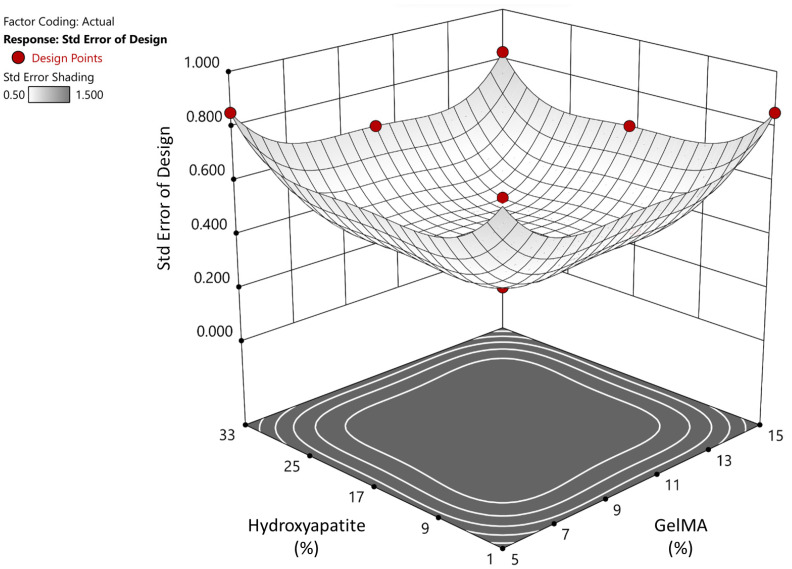
**DoE surface model**: Showing the standard error of design.

**Figure 4 materials-17-01218-f004:**
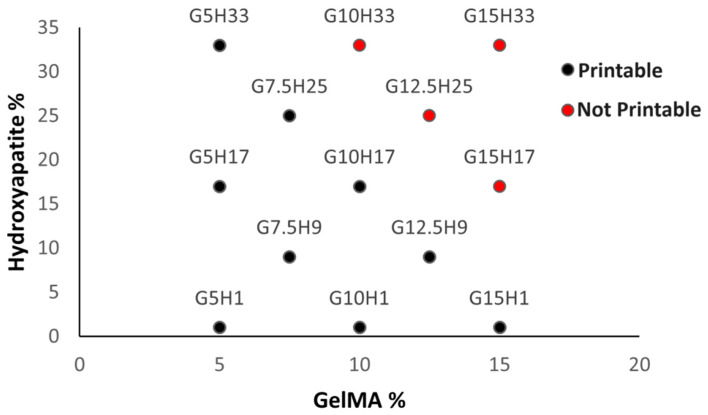
**Printability of composite hydrogels**: G10H33, G12.5H25, G15H17, and G15H33 were not printable, nor could they be pipetted under 37 °C. All other hydrogels were both extrudable and printable.

**Figure 5 materials-17-01218-f005:**
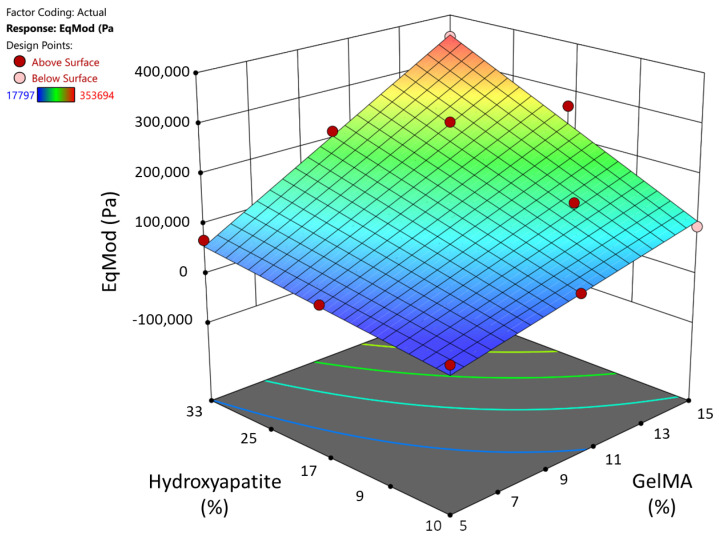
**Three-dimensional surface model of compression results**: Higher GelMA and HA % corresponded to higher equilibrium moduli.

**Figure 6 materials-17-01218-f006:**
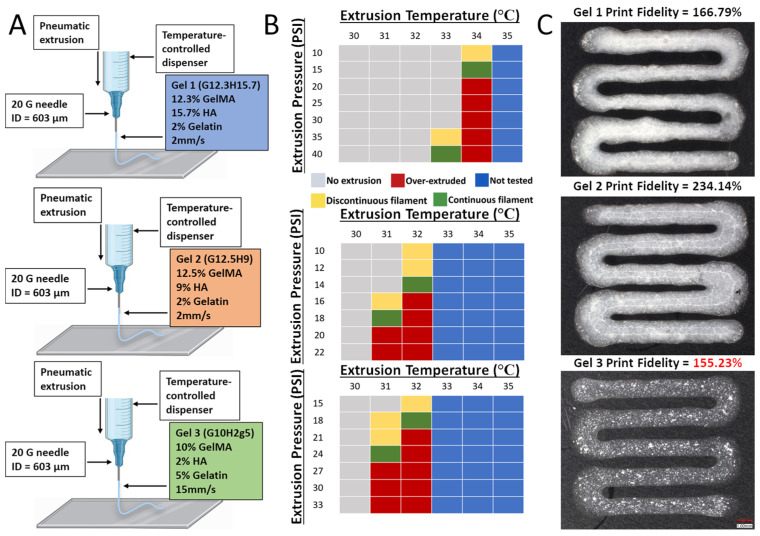
**Printability and print fidelity of GelMA/HA composite hydrogels**: (**A**) Extrusion bioprinting process and the contents of each bioink. Created using BioRender. (**B**) Determinations of print settings for extrusion of a continuous filament. (**C**) Images of printed lines with each gel (Keyence VHX-7000 microscope). Gel 3 produced the highest print fidelity. Scale bar = 1 mm.

**Figure 7 materials-17-01218-f007:**
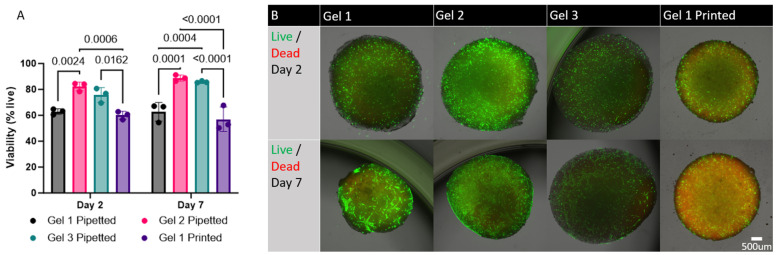
**Cell viability in GelMA/HA bioinks:** (**A**) hMSC viability in each of the three gels at day 2 and day 7. Gels 2 and 3 showed significantly higher cell viability as compared to Gel 1. There was no difference between pipetted and printed cell viability in Gel 1. There was also no difference in cell viability from day 2 to day 7 in any gel. (**B**) Live hMSCs stained green and dead hMSCs stained red; composite images are shown.

**Table 1 materials-17-01218-t001:** Initial bioink formulations.

Bioink Designation	GelMA (%)	HA (%)	Gelatin (%)
G5H1	5	1	2
G5H17	5	17	2
G5H33	5	33	2
G7.5H9A	7.5	9	2
G7.5H9B	7.5	9	2
G7.5H25	7.5	25	2
G10H1	10	1	2
G10H17A	10	17	2
G10H17B	10	17	2
G10H17C	10	17	2
G10H33	10	33	2
G12.5H9	12.5	9	2
G12.5H25A	12.5	25	2
G12.5H25B	12.5	25	2
G15H1	15	1	2
G15H17	15	17	2
G15H33	15	33	2

Bioinks with A, B, or C following them are replicates of the same composition.

**Table 2 materials-17-01218-t002:** GelMA/HA hydrogels for further testing.

	Gel 1	Gel 2	Gel 3
GelMA %	12.3	12.5	10
HA%	15.7	9	2
Gelatin %	2	2	5
Crosslinking time	60 s	60 s	60 s
Experimentally determined equilibrium modulus (kPa)	169.9	145.7	58.2

## Data Availability

The raw/processed data generated in this work are available upon request from the corresponding author.

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
