# Peer review of "Suitability of Gelatin Methacrylate and Hydroxyapatite Hydrogels for 3D-Bioprinted Bone Tissue"

_materials, 2024, doi:10.3390/ma17051218_

Round 1

Reviewer 1 Report

Comments and Suggestions for Authors

Dear Authors! Thank you very much for the study on the current issues of developing new approaches to 3D bioprinted bone tissue. 

The title of the article reflects its content. The keywords represent the same term both abbreviated and in full (DoE). One option should be removed.

Highlighting separate sections in the Abstract seems unnecessary.

Introduction 

It is not clear how the percentages relate in the statement (line 62-63): "Natural bone is typically 20-40% collagen, 60-80% mineral, and 10-20% water". For example, with a mineral content of 80%, what is the collagen and water content? (minimum values are 20 and 10, respectively, so 80+20+10=110, which is incorrect).

For the phrase (line 116-117): "Design of Experiments software (DoE) can aid in the experimental planning and analysis [13, 14, 26]" there are three references to the works of Kari Martyniak, one of the co-authors of the current paper. Apart from the redundancy of self-citation, these works are indirectly related to the above thesis. The references to primary works about software and mathematical methods of DoE are appropriate.

For the next paragraph (lines 124-128), which also contain references to the same works, keep the one that is closest to the content of the paragraph to reduce the proportion of excessive self-citation.

Otherwise, the introduction is quite informative and consistent with the study's objective.

Materials and Methods

This section is clearly and thoroughly written.

In Table 2, the accuracy of the values is different for different compositions. For the Gel 1 formulation, both the calculated parameters and those that were actually used to obtain the formulation sample should be given (the accuracy is unlikely to exceed one decimal place). The component content values should be given with uniform accuracy: for example, if the GelMA content = 12.5%, the gelatin content should be given as 2.0%. In addition, the accuracy of the Equilibrium Modulus seems excessive, the error should be noted (and the Equilibrium Modulus values should be given with the accuracy of the error, for example, 169.9(0.1) ).

Discussion and Conclusion

This section needs to be substantially revised and expanded. The authors should more clearly indicate the novelty of the obtained results (it is not obvious in the current presentation), make a comparison with similar works, both in terms of compositions and functionality (the discussion should not be limited to GelMA/HA-materials). 

Editorial remarks:

Please use a space between the characteristic value and its dimension (145 kPa, 37 °C, 8.9 mm, 200 mW etc. elsewhere (including in the captions under the figures)).

Figure 3: please improve the quality of the image

Figure 4: the frame is redundant, the dimension must be enclosed in brackets (as it is done in the other figures)

In Figures 3 and 4, the heading at the top of the figure is redundant.

Line 417-419: check the font type.

Comments on the Quality of English Language

The quality of English needs to be carefully checked throughout the whole text. For example, the sentence from the abstract (line 27-29): "This study demonstrates it is possible to create printable bone-like grafts from GelMA and HA with increased HA levels, favorable mechanical properties (145kPa) and >80% cell viability." better and correctly sounds as: "This study demonstrates that it is possible to produce printable bone-like grafts from GelMA and HA with increased HA content, favourable mechanical properties (145 kPa) and >80% cell viability."

Reviewer 2 Report

Comments and Suggestions for Authors

In this manuscript, the authors utilize DoE to explore a range of formulations for polymeric/ceramic bioink in 3D printing applications. They claimed that a gel composed of 12.5% GelMa, 9% HA, and 2% Gelatin could be the most ideal candidate for further application However, the manuscript lacks several crucial experiments. I would like to recommend a major revision before acceptance.

1.     Abstract: The study's objective or hypothesis is not clearly presented in the abstract.

2.     Given the use of polymeric/ceramic bioink for 3D printing, a comprehensive assessment of the bioink's printability is essential. This should include an analysis of the rheological properties, particularly focusing on shear-thinning behavior, rather than a simplistic binary scoring system. Authors can refer these works: doi.org/10.1016/j.cej.2021.128541; 10.1039/C3BM00012E.

3.     Can the authors provide a digital image of the printed scaffolds?

4.     How about the Hap distribution in the printed scaffold? Please provide SEM for demonstration.

5.     What’s the size information of the printed scaffold used for mechanical tests and viability?

6.     Conclusion. please state the gel composition identified as 'Gel 1' in the section

Comments on the Quality of English Language

Minor editing of English language required

Reviewer 3 Report

Comments and Suggestions for Authors

This is an interesting study and the authors have collected a unique dataset using cutting edge methodology. The paper is generally well written and structured. However, in my opinion the paper has some shortcomings in regards to some data analyses and text. Below I have some suggestions for the revision:

1.    1. It is better to have scale bars in images of Figure 2; which image is A and which one is B? What does the purple line mean?

2.    2. It is not necessary to post images of Gel 1, Gel 2, Gel 3, and Gel 1 Printed in Supplemental Figure 1 repeatedly;

3.    3. Please double check the font of lines 417 – 419.

Reviewer 4 Report

Comments and Suggestions for Authors

The paper evaluates the suitability of gelatin methacrylate (GelMA) and hydroxyapatite (HA) hydrogels for extrusion-based 3D bioprinting (3DBP) o bone tissue implants. The use of Design of Experiments (DoE) software to help in the experimental planning and analysis proved to be effective in selecting the bioink compositions and narrowing down all potential combinations to 13 to be further tested. The bioinks were further evaluated to determine the viability of human bone-marrow-derived mesenchymal stromal cells (hMSCs) in each printable bioink after the printing process in order to determine their suitability as bone replacement implants. The results of this study demonstrate that hydrogels combining GelMA and HA are suitable materials for 3DBP of bone-like constructs and have promising future implications as 3D bioprinted bone substitutes, with 2 gels warranting further studies: a GelMA/HA bioink consisting of 12.3% GelMA, 15.7% HA, and 2% Gelatin due to having the greatest mechanical stiffness, and a hydrogel consisting of 10% GelMA, 2% HA, and 5% Gelatin due to producing the best print fidelity.

The study is well thought out and carried out logically, building on the result from the DoE software and further testing to determine the ideal hydrogel content of GelMA, HA, and Gelatin to meet the required mechanical properties of a bone replacement implant. Three compositions were selected to carry out viability studies and determine successful candidates to conduct further studies on their ability to match the hardness of physiological bones in culture and their ability to promote regeneration in animal studies.

The conclusions are consistent with the presented evidence and results.

Overall, the paper is very well written, with easy-to-follow figures that add to the importance of the study.

Some suggestions

Under the design of experimental parameters, 17 gels are said to be made with some repeated compositions to test experimental variability. However, there is no mention of this later in the text. Was there any experimental variability when making and testing these gels for printability and mechanical properties?

Consider including more information on the viability of Gel 2 and 3. The authors mentioned results were not quantifiable; why?

On page 12, lines 417-419, the font appears different from the rest of the text. 

Round 2

Reviewer 1 Report

Comments and Suggestions for Authors

Dear Authors! Thank you for the changes you have made. Most of the comments have been resolved. The discussion section can still be improved, in my opinion, but further improvements are not a critical barrier to publication. Improvements can be related to comparisons with materials of the same purpose but different compositions.

Minor typos noted:

Line 203: "405nm" should be "405 nm".

Line 312,314: sample names should start with a capital letter (Gel 1, Gel 2 etc)same occurs in the Discussion (e.g. lines 403, 406, 407) - please check throughout the text.

Otherwise, after the mentioned improvements, the article can be accepted for publication.

Reviewer 2 Report

Comments and Suggestions for Authors

Most importantly, a binary scoring system represents a subjective characterization that lacks scientific rigor for accurate identification.

Comments on the Quality of English Language

Minor editing of English language required
